# Association between Appendicitis and Incident Systemic Sclerosis

**DOI:** 10.3390/jcm10112337

**Published:** 2021-05-27

**Authors:** Kuang-Tsu Yang, James Cheng-Chung Wei, Renin Chang, Chi-Chien Lin, Hsin-Hua Chen

**Affiliations:** 1Division of Gastroenterology and Hepatology, Department of Internal Medicine, Kaohsiung Veterans General Hospital, Kaohsiung 813414, Taiwan; ktyang1104@gmail.com; 2Graduate Institute of Integrated Medicine, China Medical University, Taichung 404333, Taiwan; wei3228@gmail.com; 3Department of Allergy, Immunology and Rheumatology, Chung Shan Medical University Hospital, Taichung 40201, Taiwan; 4Institute of Medicine, College of Medicine, Chung Shan Medical University, Taichung 40201, Taiwan; 5Department of Emergency Medicine, Kaohsiung Veterans General Hospital, Kaohsiung 813414, Taiwan; rhapsody1881@gmail.com; 6Department of Recreation Sports Management, Tajen University, Pingtung 90741, Taiwan; 7Institute of Biomedical Science and Rong Hsing Research Center for Translational Medicine, National Chung Hsing University, Taichung 402, Taiwan; lincc@dragon.nchu.edu.tw; 8Department of Medical Research, China Medical University Hospital, Taichung 404333, Taiwan; 9Department of Medical Research, Taichung Veterans General Hospital, Taichung 40705, Taiwan; 10Division of Allergy, Immunology and Rheumatology, Department of Internal Medicine, Taichung Veterans General Hospital, Taichung 40705, Taiwan; 11School of Medicine, National Yang-Ming University, Taipei 11221, Taiwan; 12Institute of Biomedical Science and Rong Hsing Research Centre for Translational Medicine, Chung Hsing University, Taichung 402, Taiwan; 13Institute of Public Health and Community Medicine Research Centre, National Yang-Ming University, Taipei 11221, Taiwan; 14Department of Industrial Engineering and Enterprise Information, Tunghai University, Taichung 407224, Taiwan; 15Division of General Internal Medicine, Department of Internal Medicine, Taichung Veterans General Hospital, Taichung 40705, Taiwan

**Keywords:** appendicitis, systemic sclerosis, nationwide, population-based, case–control study

## Abstract

**Objective:** This nationwide study aimed to investigate the association between newly diagnosed systemic sclerosis (SSc) and previous appendicitis history. **Methods:** A total of 1595 patients who were newly diagnosed with SSc were recruited as the SSc cases from the 2003 to 2012 claims data of the entire population in Taiwan. The other 15,950 individuals who had never been diagnosed with SSc during 2003 and 2012 were selected as the non-SSc controls to match the SSc cases. We defined that the index date as the first date of SSc diagnosis of SSc cases and the first date of ambulatory visit for any reason of non-SSc controls. Conditional logistic regression analysis was applied for the association between appendicitis and the risk of the incident SSc, tested by estimating odds ratios (ORs) with 95% confidence intervals (CIs). Potential confounders, including the Charlson comorbidity index (CCI), a history of periodontal disease, salmonella infection, and intestinal infection, were controlled. We further designed sensitivity analyses by varying the definition of appendicitis according to the status of receiving primary appendectomy. **Results:** The mean age was 51 years in the case and control groups. Females accounted for 77.5%. A total of 17 (1.1%) out of 1595 SSc cases and 81 (0.5%) out of 15,950 non-SSc controls had a history of appendicitis before the index date had a history of appendicitis. A significant association between appendicitis and the risk of SSc was confirmed (OR, 2.03; 95% CI, 1.14–3.60) after adjusting potential confounders. CCI ≥ 1 (OR, 8.48; 95% CI, 7.50–9.58) and periodontal disease (OR, 1.55; 95% CI, 1.39–1.74) were also significantly associated with the risk of SSc. The association between appendicitis and SSc risk remained robust using various definitions of appendicitis. **Conclusion:** Our study demonstrated appendicitis was associated with the incident SSc. CCI ≥ 1 and periodontal disease also contributed to the risk of developing SSc.

## 1. Introduction

Systemic sclerosis (SSc), also known as scleroderma, is a multidisciplinary autoimmune disease [1]. SSc is characterized by vascular and immune dysfunction, leading to fibrosis that can damage multiple organs [2]. SSc is divided into two subsets: limited systemic sclerosis (lcSSc) and diffuse systemic sclerosis (dsSSc). LcSSc leads to fibrosis of the skin of the face and limbs (distal to the knees and elbows), and dsSSc brings about fibrosis of the trunk and proximal parts of the limbs [3]. A 2018 global update on the epidemiology of SSc stated overall incidence rates of 8 to 56/1,000,000 person-year and prevalence rates of 38 to 341/1,000,000 person-year [4]. Furthermore, a recent systematic review and meta-analysis revealed that the standardized mortality ratio of SSc reached a high value of 3.45 [5]. SSc is still a great life-threatening disease to patients and brings numerous challenges to clinicians.

For early recognition of probable SSc development, plenty of risk factors of SSc were identified in previous research. A family history of SSc can contribute to a 13-fold higher risk of developing SSC in the patients’ first-degree relatives and 15-fold higher in patients’ siblings than that in the general population [6]. Genes such as BANK1, BLK, CD226, etc., are significantly associated with SSc [6]. In females, SSc is approximately 8–9 times more frequent than in males, but male patients will encounter a worse prognosis [7]. Environmental pathogenic factors involving silica, solvents (white spirit, trichloroethylene, aromatic solvents, chlorinated solvents, ketones, and welding fumes) [8], and heavy metals (antimony, cadmium, lead, mercury, molybdenum, palladium, and zinc) have been proven risk factors of SSc [9]. Though many efforts were made on the investigation of identifying risk factors of SSc, there are still many areas not found in correlation with SSc development.

The appendix has been long thought to be an evolutionary remnant of little significance to normal physiology, but it has more recently been identified as an important component of mammalian mucosal immune function, particularly B-lymphocyte-mediated immune responses and extrathymically derived T-lymphocytes. It also serves as a place for further development of T and B cell areas of follicles within the lymphoid tissue [10,11]. The biofilm in the appendix is thought to act as a “safe house” for commensal bacteria under the assistance of secretory IgA (sIgA) and mucin by increasing adhesive growth of the agglutinated gut flora [11]. Many studies have suggested that appendicitis or appendectomy may be associated with numerous diseases, such as ulcerative colitis [12], Crohn’s disease [13,14], and Parkinson’s disease [15]. However, the association of appendicitis or appendectomy with SSc still remained unaddressed, and there lacks a large population-based study before.

To solve this knowledge gap, we launched a nationwide, population-based, case–control study to investigate the association between appendicitis or appendectomy and SSc in Taiwan.

## 2. Materials and Methods

### 2.1. Ethics Statement

This study was approved by the Institutional Review Board of Taichung Veterans General Hospital (TCVGH CE14149B-1). This is a retrospective study, and all identified patients were anonymous. Therefore, there was no need for obtaining informed patient consent.

### 2.2. Data Source

The National Health Insurance (NHI) Program was implemented on 1 March 1995, and it has since covered more than 99% of the population in Taiwan. The National Health Insurance Research Database (NHIRD), derived from the NHI Program in 1997, includes patient characteristics (such as age, gender, and date of birth), dates of admission and discharge, the International Classification of Diseases, Ninth Revision, Clinical Modification (ICD-9-CM) codes of diagnoses and procedures, medication prescriptions, and medical expenditure covered by the NHI. The cases were selected from the whole population and controls from the Longitudinal Health Insurance Database 2000 (LHID 2000), with claims data (from 1997 to 2013) of 1 million beneficiaries. We acknowledged that the LHID 2000 database had the representative power of the national population [16].

### 2.3. Study Design

This is a nationwide, population-based, case–control study. SSc is defined as ICD-9-CM Code 710.1. From 2006 to 2012, we extracted subjects from the whole population autoimmune database who had a diagnosis of SSc with at least three outpatient visits or one inpatient visit and a catastrophic illness certificate of SSc. Patients with major or catastrophic illnesses (e.g., cancer, cirrhosis, heart failure, or autoimmune diseases) in Taiwan are recruited into the NHI registry system and get catastrophic illness certificates exempt from copayment. A routine validation of diagnoses is performed by at least two specialists from the Bureau of NHI (BNHI). They carefully review the original medical history, laboratory data, images, and pathology of all patients with catastrophic illness registration. Catastrophic illness certificates are delivered by the BNHI to those who had the major illness criteria. The database does not involve laboratory data and radiographic images, but the BNHI routinely reviews and edits diagnoses accuracy by randomly sampling charts of a patient, which will enhance coding accuracy.

Additionally, we chose individuals from the LHID with ambulatory records who never had a diagnosis of SSc during 2006 and 2012 and were matched with SSc cases (1:10) for age, sex, and the index date as non-SSc controls. The flowchart of patient enrollment is shown in Figure 1. The index date was defined as the first date of SSc diagnosis in the SSc group and the first date of ambulatory visit for any reason in the control group.

### 2.4. Definition of a History of Appendicitis

Subjects were considered to have a history of appendicitis if they had at least one ambulatory visit or inpatient visit with a diagnosis of appendicitis (ICD-9-CM Codes 540–543) before the index date.

### 2.5. Potential Confounders

Potential confounders included a history of periodontal disease (ICD-9-CM 523) [17], salmonella (ICD-9-CM 003) [18], and ill-defined intestinal infections (ICD-9-CM 009) [19] before the index date, as well as a Charlson comorbidity index (CCI) within one year before the index date. Deyo et al. indicated that CCI could be applied to evaluate the degree of clinically general comorbid situations [20].

### 2.6. Sensitivity Analysis

For certifying the robust association between appendicitis and SSC, we also performed sensitivity analyses by varying the definition of appendicitis according to the status of receiving primary appendectomy. Different definitions of appendicitis involved appendicitis (ICD9: 540–543), primary appendectomy (ICD9 Procedure Code: 47.0), appendicitis and primary appendectomy, and appendicitis or primary appendectomy.

### 2.7. Statistical Analysis

We compared baseline characteristics between cases and controls using Pearson’s χ^2^ test for categorical variables and a *t*-test for continuous variables. Conditional logistic regression was used to examine the influence of appendicitis and other potential confounders on the risk of SSc development, as shown by odds ratios (ORs) with 95% confidence intervals (CIs). A two-tailed value of *p* < 0.05 was considered statistically significant. All statistical analyses were performed using SAS version 9.3 (SAS Institute, Inc., Cary, NC, USA).

## 3. Results

Figure 1 shows the flowchart of patient recruitment. We identified 1595 SSc cases and 15,950 non-SSc controls. Table 1 reveals the demographic data and clinical characteristics of SSc cases and non-SSc controls. The enrolled individuals’ ages range from 4 to 91 years old. Among them, those who are over 45 years old accounted for 66.8%. The ratios of individuals with CCI more than and equal to one (59.6% vs. 19.2%), myocardial infarction (0.5% vs. 0.2%), congestive heart failure (3.4% vs. 0.7%), peripheral vascular disease (3.1% vs. 0.2%), chronic obstructive pulmonary disease (COPD) (9.8% vs. 3.0%), connective tissue disease (39.6% vs. 0.3%), peptic ulcer disease (11% vs. 4.4%), mild liver disease (5.1% vs. 2.1%), diabetes mellitus with end-organ damage (2.3% vs. 1.5%), hemiplegia (0.6% vs. 0.2%), moderate to severe renal disease (5.5% vs. 1.1%), tumor (6.6% vs. 2.1%), and metastatic solid tumor (1.0% vs. 0.2%) are more in SSc cases than in non-SSc. Similarly, appendicitis (1.1% vs. 0.5%), primary appendectomy (0.9% vs. 0.5%), appendicitis and primary appendectomy (0.9% vs. 0.5%), appendicitis or primary appendectomy (1.1% vs. 0.5%), ill-defined intestinal infections (14.4% vs. 12.7%), and periodontal disease (48.5% vs. 37.2%) all have a trend of higher ratio in SSc cases than non-SSc controls.

Our study uses a conditional logistic regression model to investigate the association of SSc risk with appendicitis, CCI, periodontal disease, salmonella, and ill-defined intestinal infections (Table 2). We found that appendicitis, CCI ≥ 1, periodontal disease, and ill-defined intestinal infections are associated with SSc significantly in univariable analysis (appendicitis, OR, 2.11; 95% CI, 1.25–3.57; CCI ≥ 1, OR = 8.64, 95% CI = 7.64–9.76; periodontal disease, OR = 1.64, 95% CI = 1.48–1.83; ill-defined intestinal infections, OR = 1.17, 95% CI = 1.004–1.35). In multivariable analysis, appendicitis was still a significant risk factor of SSc (OR, 2.03; 95% CI, 1.14–3.60). CCI ≥ 1 and periodontal disease were also risk factors of SSc.

Our sensitivity analysis for researching the association of different diagnosis definitions of appendicitis with SSc are shown in Table 3. Different diagnosis codes include appendicitis, primary appendectomy, appendicitis and primary appendectomy, and appendicitis or primary appendectomy. We discover that different diagnosis definitions of appendicitis are all significant risk factors of SSc, and the ORs are approximately two. In other words, those patients who had a history of appendicitis have a two-fold risk of suffering from SSc compared with those without a history of appendicitis.

## 4. Discussion

To our knowledge, this nationwide, population-based, case–control study firstly utilized longitudinal administrative data to examine the strength of the association between appendicitis and SSc. The main finding of this study was a statistically significant association between the history of appendicitis and the risk of incident SSc after adjustment for periodontal disease, CCI, salmonella, and ill-defined intestinal infection. The association between appendicitis and SSc risk is still robust using various definitions of appendicitis correlated with the status of primary appendectomy.

The precise mechanism between appendicitis SSc was not well established currently. After the literature review, we propose two hypotheses that might explain this phenomenon. The first hypothesis is that the appendix has an immunologically regulatory role. T cells in the lamina propria of the appendix express more of the integrin subunit β_7_ compared to T and B cells throughout other sites of the gut. Two kinds of integrin-α_4_β_7_ and α_E_β_7_ interact with lymphocytes. Of note, α_E_β_7_ is capable of stimulating differentiation forkhead box protein 3 (FoxP3)^+^ T_reg_ cells. As long as this differentiation is suppressed, a proinflammatory state could occur [11]. Subsequent proinflammatory cytokines (type I interferons, TGF-β, interleukin 17, and interleukin 23) and mediators (leptin, adiponectin, chimerin, and interleukin 6) would promote the pathogenesis of SSc [21].

The second hypothesis is that intestinal microbiota change was considered a possible pathogenetic factor of SSc after appendectomy or appendicitis. Research has shown SSc patients had significantly higher levels of pathobiont genera, such as *Fusobacterium, Prevotella*, and *Proteobacteria* genera, compared with controls [22]. *Fusobacterium* species belong to the Gram-negative anaerobes category and appear mostly in the oral cavity. If they are present in the colon, they will show their invasive ability and translocate into the systemic circulation. They could induce bacteremia, organ abscesses, possibly coronary artery disease [23,24], and inflammatory bowel disease. *Fusobacterium* species could bring about more invasive and proinflammatory properties in cultured epithelial cell assays than those strains isolated from healthy individuals, which implies that *Fusobacterium* species are linked with inflammatory diseases [25]. *Prevotella* species were more abundant in SSc patients than healthy people. These genera are increased in patients with Crohn’s disease or rheumatoid arthritis [26,27]. A study had also demonstrated that there was a significantly increased level of Proteobacteria in patients with Crohn’s disease [28]. Mucosal biopsies showed a significantly higher level of Proteobacteria in patients with extensive and active ulcerative colitis than in patients with a limited extent and less active disease [29]. Some studies reported the normal human appendix harbors populations of *Fusobacterium*, *Prevotella*, and *Proteobacteria* genera that are generally absent in fecal samples from healthy adults and children [30,31]. After appendicitis or appendectomy, *Fusobacterium, Prevotella*, and *Proteobacteria* genera will lose their harbor and may initiate the inflammation cascade and the subsequent development of SSc.

Our results demonstrated that periodontal disease was the risk factor of SSc development. The potential mechanism might be explained by the association between *Prevotella*, inflammation, and periodontitis [32]. *Prevotella* was confirmed in biofilms of gingivitis and periodontitis [33] and was well identified as a driver of neutrophil recruitment, proinflammatory cytokines, and metalloproteinase expression attributing destruction of connective tissues and alveolar bone [34]. Prevotella nigrescens in mice can induce periodontal disease and promote immune responses such as increasing T helper type 17 (Th17) (i.e., interleukin (IL-17)) and Th1 (interferon-γ (IFN-γ)) cytokine production by lymph node T cells compared with uninfected mice [35]. IL-17 was found capable of indirectly enhancing the fibrotic process in experimental animal models of skin and lung fibrosis. IL-17 could induce inflammation by recruiting inflammatory cells, stimulating the production of TGF-β and other profibrotic mediators, and inhibiting autophagy [36]. On the other hand, serum IFN-γ levels were higher in SSc patients than in individuals without SSc. Furthermore, IFN-γ is associated with pulmonary arterial hypertension and positively correlated with mean systolic pulmonary arterial pressure, which suggested its role in pulmonary manifestations of SSc patients [37].

Salmonella infection was considered as a potential risk factor of SSc in our study based on the previous research [18], which indicated that nontyphoidal salmonella bacteremia causes substantial morbidity and mortality in patients with connective tissue disease, including scleroderma. However, in our study, the final results showed there was no significant association between salmonella infection and SSc. Such inconsistency may be explained by the fact that we considered salmonella infection as a confounder but not salmonella bacteremia, which suggested a more severe form of salmonella infection that might lead to a greater influence on SSc risk.

CCI ≥ 1 was also significantly associated with the risk of developing SSc in our study. We hypothesized that before the index date, patients with SSc would first have constitutional symptoms and signs, including cutaneous manifestations (e.g., pruritus, edema, skin hyperpigmentation), gastrointestinal involvement (e.g., dysphagia, choking, heartburn sensation, hoarseness, bloating), pulmonary involvement (e.g., breathlessness on exertion, nonproductive cough), cardiac manifestations (e.g., chest pain), renal involvement (e.g., impaired renal reserve, hypertension, microalbuminuria), and neuromuscular involvement (e.g., muscle atrophy, weakness, myopathy, neuropathy). Due to the above-mentioned symptoms and signs, patients were easily diagnosed with other comorbidities before the diagnosis of SSc. Therefore, CCI ≥ 1 was also identified as a risk factor of SSc in our study.

The strength of our study is using population-based data to minimize selection bias. However, we acknowledged that there are some limitations of our study. First, the NHIRD does not include personal information regarding lifestyles, body mass index values, smoking, and alcohol use. Smoking currently is still not clarified as a certain risk factor of SSc [38]. In order to minimize these biases, we had adjusted COPD and comorbidities to proxy these lifestyle-related diseases. Second, patients’ laboratory parameters and family history were not available in this dataset, which are also confounders of the development of SSc. Third, coding errors might happen in the NHIRD. To alleviate the bias, we recruited patients under the criteria of a diagnosis of SSc with at least three outpatient visits or one inpatient visit and a catastrophic illness certificate of SSc. Finally, our results may not be able to be generalized to non-Taiwanese populations.

## 5. Conclusions

Our nationwide, population-based, case–control study demonstrated an association between appendicitis and the risk of SSc. Further mechanistic studies are warranted to elucidate the causal relationship between appendicitis and SSc development.

## Figures and Tables

**Figure 1 jcm-10-02337-f001:**
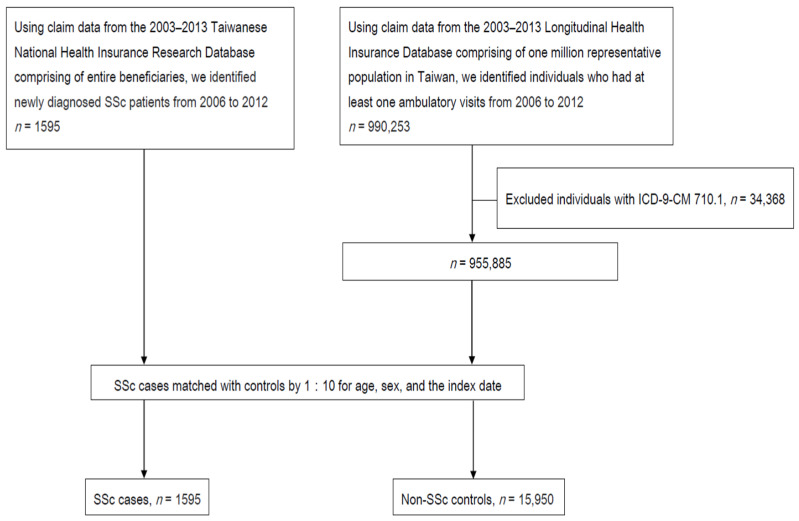
Study flowchart.

**Table 1 jcm-10-02337-t001:** Demographic data and clinical characteristics between the two study groups.

	Control	Case	
	(*n* = 15,950)	(*n* = 1595)	*p*-Value
Age, years (mean ± SD)	51 ± 15	51 ± 15	1
Gender			1
Female	12,360 (77.5)	1236 (77.5)	
Male	3590 (22.5)	359 (22.5)	
Appendicitis	81 (0.5)	17 (1.1)	0.004
Primary appendectomy	74 (0.5)	15 (0.9)	0.011
Appendicitis and primary appendectomy	73 (0.5)	15 (0.9)	0.009
Appendicitis or primary appendectomy	82 (0.5)	17 (1.1)	0.005
Salmonella	65 (0.4)	8 (0.5)	0.578
Ill-defined intestinal infections	2020 (12.7)	230 (14.4)	0.047
Periodontal disease	5936 (37.2)	774 (48.5)	<0.001
CCI group			<0.001
0	12,888 (80.8)	645 (40.4)	
≥1	3062 (19.2)	950 (59.6)	
Myocardial infarction	24 (0.2)	8 (0.5)	0.002
Congestive heart failure	109 (0.7)	55 (3.4)	<0.001
Peripheral vascular disease	36 (0.2)	50 (3.1)	<0.001
Cerebrovascular disease	439 (2.8)	52 (3.3)	0.241
Dementia	75 (0.5)	5 (0.3)	0.376
COPD	485 (3.0)	156 (9.8)	<0.001
Connective tissue disease	53 (0.3)	631 (39.6)	<0.001
Peptic ulcer disease	700 (4.4)	175 (11)	<0.001
Mild liver disease	332 (2.1)	82 (5.1)	<0.001
Diabetes mellitus	1158 (7.3)	113 (7.1)	0.797
Diabetes mellitus with end-organ damage	246 (1.5)	37 (2.3)	0.019
Hemiplegia	35 (0.2)	10 (0.6)	0.002
Moderate to severe renal disease	181 (1.1)	87 (5.5)	<0.001
Tumor	341 (2.1)	105 (6.6)	<0.001
Moderate or severe liver disease	13 (0.1)	3 (0.2)	0.179
Metastatic solid tumor	34 (0.2)	16 (1.0)	<0.001
AIDS	1 (0)	0 (0)	0.752

Abbreviations: SD, standard deviation; CCI, Charlson comorbidity index; COPD, chronic obstructive pulmonary disease; AIDS, acquired immune deficiency syndrome.

**Table 2 jcm-10-02337-t002:** Conditional logistic regression analyses for risk factors of SSc.

	Univariate Analysis	Multivariate Analysis
	OR (95% CI)	OR (95% CI)
Diagnosed with appendicitis	2.11 (1.25–3.57)	2.03 (1.14–3.60)
CCI ≥ 1	8.64 (7.64–9.76)	8.48 (7.50–9.58)
Periodontal disease	1.64 (1.48–1.83)	1.55 (1.39–1.74)
Salmonella	1.23 (0.59–2.58)	0.97 (0.43–2.18)
Ill-defined intestinal infections	1.17 (1.004–1.35)	1.00 (0.85–1.17)

Abbreviations: SSc, systemic sclerosis; CCI, Charlson comorbidity index.

**Table 3 jcm-10-02337-t003:** Sensitivity analysis.

	Control (*n* = 15,950)		Case (*n* = 1595)		Multivariate Analysis
	Event (%)	Duration (Year) (Mean ± SD)	Event (%)	Duration (Year) (Mean ± SD)	OR (95% CI)
Appendicitis	81 (0.5)	3.7 ± 2.2	17 (1.1)	3.7 ± 2.1	2.03 (1.14–3.60)
Primary appendectomy	74 (0.5)	3.6 ± 2.1	15 (0.9)	3.7 ± 2.1	1.93 (1.06–3.54)
Appendicitis and Primary appendectomy	73 (0.5)	3.6 ± 2.1	15 (0.9)	3.7 ± 2.1	1.97 (1.07–3.61)
Appendicitis or Primary appendectomy	82 (0.5)	3.7 ± 2.2	17 (1.1)	3.7 ± 2.1	1.99 (1.13–3.53)

## Data Availability

Data are available from the National Health Insurance Research Database (NHIRD) published by Taiwan National Health Insurance (NHI) Bureau. Due to legal restrictions imposed by the government of Taiwan in relation to the “Personal Information Protection Act”, data cannot be made publicly available.

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
