# Peer review of "Association between Appendicitis and Incident Systemic Sclerosis"

_jcm, 2021, doi:10.3390/jcm10112337_

Round 1

Reviewer 1 Report

Comments to Authors

In the manuscript „Association between appendicitis and incident SSc” the authors proved that the appendicitis was associated with the incident SSc and periodontal disease also contributed to the risk of developing SSc. The observations presented by the authors are very interesting and useful in clinical practise. I am impressed with inclusion of such a large study and control group. The quality of study are worth stressing, the worth of the study is increased by adding control group. The manuscript presents very interesting results,

It could be worth presenting more information about study groups.

To sum up, I would like to congratulate on this extremely difficult and high-level work. I suggest continuing research.

Your sincerely

Author Response

Thank you for considering our manuscript for further revisions.

We agreed that more information about the study groups would be needed. We had added our detailed information about CCI (Supplementary Table, Page 4, line 159-160) to reveal the differences between the two study groups.

We will continue further work about this issue. Thank you very much!

Reviewer 2 Report

In this case-control population-based study, the Authors report a two-fold association between previous appendicitis and incident diagnosis of systemic sclerosis (SSc).

Overall, the study is well designed and written. The methods are accurate, and the limitations of the study are those of all case-control designs.

In order to minimize selection and ascertainment biases, the Authors used a nationwide database to extract data of 1595 SSc patients, matched 1:10 with non-SSc subjects. They adjusted the analysis for potential cofounders, such as periodontal disease, salmonella, ill-defined infections and comorbidities. However, additional potential confounders could not be measured. The Authors correctly acknowledged these potential biases in the Discussion.

Statistical analysis is robust as sensitivity analyses were also conducted.

English needs improvement and spell checks.

Author Response

Thank you for considering our manuscript for further revisions.

We had corrected some grammar and vocabulary errors. 

Reviewer 3 Report

 This Taiwanese retrospective case-control study demonstrated appendicitis was associated with the incident SSc. Charlson comorbidity index (CCI) ≥1 and periodontal disease also contributed to the risk of developing SSc. 1595 patients who 32 were newly diagnosed SSc and the other 15950 individuals who had never been diagnosed SSc were included. The number of this study is large. A significant association between appendicitis and the risk of SSc was confirmed. CCI ≥1 and periodontal disease were also significantly associated with the risk of SSc. The association between appendicitis and SSc risk remained robust using various definitions of appendicitis.

 There was no study to reveal the association between appendicitis and SSc. This study is interesting. The strong point of this study is large number of patients.
In this study, it is not clear how common appendicitis is in SSc patients. It is necessary to confirm this by interviewing actual patients and setting up a cohort or registry of post-appendicitis patients.

 To make use of this study in the future, it is also of concern whether something can be done to suppress the onset of SSc in patients with post appendicitis or periodontal disease. All in all, we believe that the results are worthy of publication because the point of view is new and the results are interesting.

Author Response

Thank you for considering our manuscript for further revisions.

We acknowledged that further actual patients interviewing is needed to confirm the details in our study. However, due to the limitation of the NHIRD and the anonymous data of study groups acquired, we were not able to find the actual patients enrolled in our study. We could only get their basic characteristics, diagnoses, and other comorbidities. 
